# Preparation and Characterization of Calcium-Chelated Sea Cucumber Ovum Hydrolysate and the Inhibitory Effect on α-Amylase

**DOI:** 10.3390/foods13244119

**Published:** 2024-12-20

**Authors:** Xu Yan, Fengjiao Fan, Zijin Qin, Lijuan Zhang, Shuang Guan, Shiying Han, Xiufang Dong, Hui Chen, Zhe Xu, Tingting Li

**Affiliations:** 1Key Laboratory of Biotechnology and Bioresources Utilization, College of Life Sciences, Dalian Minzu University, Ministry of Education, Dalian 116600, China; yx13752409672@163.com (X.Y.); 20231575@dlnu.edu.cn (L.Z.); 19818957899@163.com (S.G.); 19853500716@163.com (S.H.); 2College of Food Science and Engineering, Nanjing University of Finance and Economics, Nanjing 210023, China; fanfjklyx@163.com; 3Department of Food Science and Technology, The University of Georgia, Athens, GA 30602, USA; zq20739@uga.edu; 4College of Marine Science and Biological Engineering, Qingdao University of Science and Technology, Qingdao 266042, China; dxf900321@126.com; 5Key Laboratory of Marine Fishery Resources Exploitment & Utilization of Zhejiang Province, Zhejiang University of Technology, Hangzhou 310014, China; realcrital@126.com

**Keywords:** sea cucumber ovum, hydrolysate, α-amylase, inhibitory activity, calcium chelation

## Abstract

α-amylase can effectively inhibit the activity of digestive enzymes and alter nutrient absorption. The impact of ovum hydrolysates of sea cucumbers on α-amylase activity was investigated in this study. The protein hydrolysates generated using different proteases (pepsin, trypsin, and neutral protease) and molecular weights (less than 3000 and more than 3000) were investigated. The results showed that all three different hydrolysates demonstrated calcium-chelating activity and induced a fluorescence-quenching effect on α-amylase. The sea cucumber ovum hydrolysate with a molecular weight of less than 3000 Da, isolated using trypsin, showed the most effective inhibitory effect on α-amylase, with an inhibition rate of 53.9%, and the inhibition type was identified as mixed forms of inhibition. In conclusion, the generation and utilization of protein hydrolysates from sea cucumber ovum as a functional food ingredient could be a potential approach to add value to low-cost seafood by-products.

## 1. Introduction

The prevalence of chronic diseases such as obesity and diabetes are significantly associated with the consumption of a Western diet, which is characterized by a high intake of refined sugars and fat [1,2,3]. The excessive consumption of refined carbohydrates leads to a high glycemic response, triggering increased insulin secretion, which further promotes enhanced fat storage. Both processes are major contributors to obesity [4,5]. Carbohydrate digestion begins in the oral phase with salivary amylase. The intestinal pancreatic α-amylase further breaks down polysaccharides into short-chain molecules including dextrin and oligosaccharides. Subsequently, these short-chain molecules are hydrolyzed to glucose, which is absorbed by the intestinal epithelium and serves as a main energy source. A key strategy for managing diabetes is the regulation of postprandial glucose levels, and inhibition of α-amylase and α-glucosidase is one of the therapeutic approaches [6]. Most α-amylases contain at least one calcium ion, which is essential for maintaining the enzyme’s stability and activity [7,8,9]. The removal of calcium ions has been shown to result in the irreversible inactivation of α-amylase; thus, calcium chelators may have potential as α-amylase inhibitors.

The edible aquatic animals, particularly by-products such as seafood, have considerable potential for application as functional food ingredients [10]. The reuse of specific components of aquatic organisms, such as proteases extracted from fish offal, provides a sustainable practice in biotechnology and the food industry [11]. The class of marine echinoderms known as sea cucumbers contain a variety of nutrients and bioactive compounds, making them popular aquaculture products in Asian countries. For several centuries, sea cucumbers have been recognized and consumed as a food supplement due to a range of physiological benefits, including cancer prevention, anti-aging effects, and blood pressure reduction, etc. [12]. Currently, the comprehensive utilization of the by-products generated during the processing of sea cucumbers is relatively low, and the resources in their ova have yet to be fully exploited, particularly the abundant protein resources present. These may contain a multitude of biologically active peptide sequences, and it would be beneficial for researchers to conduct further in-depth and systematic research on them [13,14,15].

Bioactive peptides derived from natural sources show multiple advantages, including safety, good digestibility, and reduced irritation effects [16,17]. Peptides from aquatic organisms, particularly sea cucumbers, demonstrate significant potential for application in dietary interventions [18]. Sea cucumbers, a commercially marine organism in Asian countries, are a rich source of proteins, lipids, and minerals and have demonstrated a range of bioactive properties, including anticancer, anti-aging, and blood-pressure-lowering effects [19]. In comparison to individual amino acids or proteins, low-molecular-weight peptides exhibit heightened biological activity due to their reduced osmotic pressure and enhanced absorptive capacity [20]. Sommella et al. employed ultrafiltration and solid-liquid extraction to isolate and purify components with a molecular weight below 3000 Da and subsequently identified their biological activities, which included antioxidant, immunomodulatory, antibacterial, and antihypertensive activities [21].

This study examines the inhibitory impacts of calcium-binding peptides obtained from sea cucumber ovum on α-amylase. By investigating the calcium ion binding site of α-amylase, these peptides may provide a novel approach to weight control. Furthermore, the findings of this study may facilitate the alternative utilization of aquatic by-products, thereby contributing to the sustainable exploitation of resources.

## 2. Materials and Methods

### 2.1. Materials and Chemicals

Sea cucumber ova were obtained from Dalian Foods Co. (Dalian, China); α-amylase from porcine pancreas (10 U/mg, tested by 1% soluble starch as substrate) and soluble starches were sourced from Shanghai Yuanye Biotechnology Co., Ltd. (Shanghai, China); and acetonitrile (ACN) was procured from Merck KGaA Co. (Darmstadt, Germany). In this study, all other reagents and chemicals were of analytical grade and of the highest quality, as defined by the relevant standard.

### 2.2. Preparation of Sea Cucumber Ovum Hydrolysate and the Calcium Complex Form

Three parts of sea cucumber ovum were added into one part of distilled water, and the mixture was homogenized into a pulp (ANgni Instruments Co., Shanghai, China). The mixture was continuously stirred at 4 °C for 4 h followed by centrifugation (10,000× *g* for 10 min at 4 °C) to collect the supernatant. This process was repeated two times, and the final supernatant was freeze-dried to obtain water-soluble protein from the sea cucumber ovum. The freeze-drying method is employed to preserve the integrity and activity of the protein while facilitating its storage and further analysis.

To prepare sea cucumber ovum protein hydrolysates, pepsin, trypsin, and neutral protease were employed to hydrolyze sea cucumber ovum proteins due to their distinct pH optima and substrate specificities, allowing for a comprehensive comparison of the resulting hydrolysates’ properties. A total of 2 g of water-soluble protein was prepared by dissolving the substance in 100 mL of distilled water. The enzymatic digestion process was conducted at the optimal temperature and pH conditions under constant stirring maintained for a period of five hours (pepsin, pH 2.0, 37 °C; trypsin, pH 8.0, 37 °C; neutral protease pH 7.0, 50 °C). During digestion, the optimal pH was maintained by adding 1 M NaOH or HCl. At the end of the enzymatic digestion, the pH was adjusted to neutral, and the enzyme was inactivated by boiling the sample for 10 min. After being cooled to room temperature, the mixture was centrifuged at 10,000× *g* for 10 min at 4 °C, and the supernatant was freeze-dried to obtain sea cucumber ovum protein hydrolysates [22,23].

The calcium complexes were prepared [24]. Briefly, 5 mg of lyophilized hydrolysate powder was dispersed into with 1 mL of 5 mM CaCl_2_ and 2 mL of phosphate-buffered solution (PBS, 20mM, pH = 7), and the pH was adjusted to 8, which is the optimum pH for maintaining the stability of the calcium complex. The mixed solution was stirred continuously for one hour at a temperature of 50 °C and a pH of 8.0 [25]. Subsequently, anhydrous ethanol was introduced to achieve a final concentration of 90%. The mixture was subjected to centrifugation at 10,000× *g* for 10 min at 4 °C, and the precipitate was collected and freeze-dried to obtain peptide–calcium complex, labelled as SCOPH-Ca, SCOTH-Ca, and SCONPH-Ca, respectively.

### 2.3. Determination of the Molecular Weight Distribution

For molecular weight analysis, samples at a concentration of 1 mg/mL were filtered through a 0.45 μm polyethersulfone membrane (Guangzhou Jet Bio-Filtration Co., Guangzhou, China) and analyzed via liquid chromatography (Agilent Technologies, Santa Clara, CA, USA) [26].

### 2.4. Characterization of Sea Cucumber Ovum Hydrolysates and the Calcium Complex

#### 2.4.1. UV–Visible Spectroscopy

To evaluate the molecular structure, the samples were subjected to analysis using a UV-Vis spectrometer (UV-2450, Shimadzu, Kyoto, Japan), with ultrapure water serving as the blank control. The scanning was carried out at a rate of 1200 nm/min within a wavelength range of 200 to 800 nm.

#### 2.4.2. Fluorescence Spectroscopy

The samples were subjected to analysis using a fluorescence spectrometer (LS55, Perkin Elmer, Waltham, MA, USA), with the scanning range set to encompass the wavelength range from 310 to 500 nm. Excitation took place at a wavelength of 295 nm with a slit width of 10 nm.

#### 2.4.3. Fourier-Transform Infrared Spectroscopy

For 2 mg of powder sample was mixed with 200 mg of potassium bromide powder (KBr) for FTIR analysis. Pure KBr being used as a blank background. The samples were scanned using an FTIR spectrometer (IR Prestige-21, Shimadzu, Japan) over the wavelength range of 4000 to 400 cm^−1^.

#### 2.4.4. Particle Size Distribution

The particle size of the 1 mg/mL samples dispersed in ultrapure water was determined by a dynamic light scattering particle size meter (SZ-100, Horiba, Kyoto, Japan).

#### 2.4.5. Circular Dichroism Spectrum

The secondary structure of the sample with a concentration of 0.5 mg/mL was analyzed via circular dichroism spectrometry (Chirascan V100, Applied Photophysics, Leatherhead, Surrey, UK). The wavelength range was set to 190 to 260 nm, the slit width to 0.2 nm, and the cuvette path length to 1 mm. The experimental results were obtained by subtracting the buffer background from the scanned data [27].

#### 2.4.6. Calcium-Binding Capacity

In order to evaluate the calcium-binding capacity, a 5 mg sample was combined with 1 mL of 5 mM CaCl_2_ and 2 mL of PBS (20 mM, pH 7.0). The mixture was then incubated at 37 °C for one hour to permit calcium binding. The unbound calcium was collected by centrifugation (2652× *g* for 10 min), and the calcium content of the resulting supernatant was determined via the colorimetric method using methyl thymol blue. The difference in calcium content before and after binding was used to calculate the calcium-binding capacity according to Equation (1):(1)$$Calcium-binding\;capacity(\%)=\left[\left(C_{0}-C\right)/C_{0}\right]\times 100\%$$

In this equation, C_0_ represents the absorbance of the control solution, and C denotes the absorbance of the solution following calcium binding.

### 2.5. Pre-Separation of Enzymatic Hydrolysates of Sea Cucumber Ovum Trypsin Hydrolysate (SCOTH)

SCOTH powder was dissolved in deionized water and then separated using Amicon Ultra-4 3 KDa cutoff centrifugal filter units (Millipore, Burlington, MA, USA). The obtained fractions with a molecular weight higher than 3000 Da were identified as long-molecular-weight hydrolysates (LHs), and the rest was classified as small-molecular-weight hydrolysates (SHs). Both LHs and SHs fractions was freeze-dried for future use.

### 2.6. Determination of the Inhibitory Effect of Sea Cucumber Ovum Hydrolysate on α-Amylase and Its Mechanism

#### 2.6.1. α-Amylase Inhibitory Activity

The activity of α-amylase was assessed through spectrophotometric analysis, which enabled the measurement of the enzyme’s catalytic efficiency. During this process, soluble starch was used as the substrate, and 3,5-dinitrosalicylic acid (DNS) acted as the chromogenic reagent. Samples of α-amylase (4 U/mL), along with SCOPH, SCOTH, and SCONPH at various concentrations (1 mg/mL, 100 μg/mL, 10 μg/mL, and 1 μg/mL), were prepared in PBS (0.2 M, pH 6.9). The mixture was incubated at 37 °C for 10 min, and, after incubation, a 1% wt starch solution was added for a 5 min reaction. Subsequently, 1 mL of DNS reagent was added, and the mixture was subjected to boiling for 10 min to terminate the reaction. The absorbance of the solution was determined at a wavelength of 540 nm using a microplate reader (Synergy H1, BioTek, Winooski, VT, USA) [28], and the inhibition rate was calculated using Equation (2) as follows:(2)$$I\%=\left[1-\left(B-b\right)/\left(A-a\right)\right]\times 100\%$$

The rate of α-amylase inhibition achieved by SCOPH, SCOTH, and SCONPH is represented by I%. The term “A” is used to denote the absorbance of the blank tube, “a” to denote that of the blank control, “B” to denote that of the inhibition tube, and “b” to denote that of the background control.

To determine the impact of the gastrointestinal digestion on the inhibitory effect, the in vitro digestion simulation was conducted according to a previous study [29]. The samples were collected at 0, 10, 30, 60, 90, 120, 130, 150, 180, 210, and 240 min during the digestion. The inhibition rates of SCOPH, SCOTH, and SCONPH for α-amylase were determined using the DNS method.

#### 2.6.2. α-Amylase Conformation

The impact of hydrolysates on the secondary structure of α-amylase was investigated through the utilization of circular dichroism spectroscopy (Chirascan V100, Applied Photophysics, UK). α-Amylase was reacted with 1 mg/mL, 100 μg/mL, 10 μg/mL, and 1 μg/mL of each of the hydrolysates, while the reactions without SCOPH, SCOTH, or SCONPH were used as controls for comparison.

#### 2.6.3. Dynamic Light Scattering (DLS)

The solutions of α-amylase and various concentrations of SCOPH, SCOTH, and SCONPH were subjected to analysis using dynamic light scattering (SZ-100, Horiba, Japan).

### 2.7. Fluorescence Spectroscopy

The fluorescence spectrum of the resulting mixture of hydrolysate solutions and α-amylase was determined as previously demonstrated in a related study, with minor modifications [30]. Briefly, the hydrolysates at different concentrations were mixed with 4 U/mL α-amylase, and the reaction was conducted at 298 K, 308 K, and 318 K for one hour. The samples were then scanned using fluorescence spectroscopy (LS55, Perkin Elmer, Waltham, MA, USA).

The excitation wavelength was maintained at 280 nanometers and the scanning range was set between 310 and 500 nm, with a slit width of 5 nm and a scanning speed of 100 nm per minute. Fluorescence quenching was calculated using the Stern–Volmer equation:(3)$$\frac{F_{0}}{F}=1+K_{q}\tau_{0}\left[Q\right]=1+K_{SV}\left[Q\right]$$

In this context, F_0_ and F represent the fluorescence intensity in the absence and presence of the hydrolysate, respectively; the variable [Q] represents the concentration of the hydrolysate; K_SV_ denotes the quenching constant; K_q_ represents the rate constant for biomolecule quenching; and τ_0_ is the average fluorophore lifetime (τ_0_ = 10^−8^ s) [31].

### 2.8. Determination of Inhibition Mechanism

Determination of the kinetics of α-amylase inhibition was carried out following a previously literature [32]. A solution of 10 mg/mL starch was combined with hydrolysates of varying concentrations (1 mg/mL, 100 μg/mL, 10 μg/mL, 1 μg/mL), and the mixture was incubated at 37 °C for 30 min. Subsequently, different concentrations of α-amylase solution (500 μg/mL, 400 μg/mL, 300 μg/mL, 200 μg/mL, 100 μg/mL) were added, and the change in absorbance was quantified in order to assess the activity of the enzyme. A graph was constructed to illustrate the relationship between the enzyme reaction rate and the concentration of α-amylase.

A solution of soluble starch at various concentrations (25 mg/mL, 20 mg/mL, 15 mg/mL, 10 mg/mL, 5 mg/mL) was added to investigate the impact of hydrolysate concentrations (1 mg/mL, 100 μg/mL, 10 μg/mL, 1 μg/mL) on α-amylase activity at 37 °C, and the inhibitory effect of the hydrolysate on α-amylase was determined using Lineweaver–Burk double reciprocal plots.

### 2.9. Statistical Analyses

The results are presented as the mean ± standard deviation. The statistical analysis was conducted using IBM SPSS Statistics 26.0 (IBM Corp., Armonk, NY, USA). The data were subjected to a one-way analysis of variance (ANOVA) followed by Duncan’s post-hoc test. The level of statistical significance was set at *p* < 0.05.

## 3. Results and Discussion

### 3.1. Molecular Weight Distribution

The molecular weight distribution of SCOPH, SCOTH, and SCONPH is shown in Figure 1. The majority of hydrolysates had a molecular weight of less than 500 Da. The average molecular weights were calculated to be 1983 Da for SCOPH, 1042 Da for SCOTH, and 1705 Da for SCONPH. The calcium-ion-chelating ability is greatly depended on the molecular weight of the hydrolysates, and hydrolysates below 3000 Da was reported of higher affinity for calcium ions [33].

### 3.2. Characterization of SCOPH, SCOTH, and SCONPH and SCOPH-Ca, SCOTH-Ca, and SCONPH-Ca

#### 3.2.1. UV Absorption Spectroscopy of SCOPH, SCOTH, and SCONPH with Calcium

Figure 2A_1_–A_3_ showed difference in UV-Vis spectrum among protein hydrolysates and their calcium complex forms. The spectrum of protein hydrolysates (SCOPH, SCOTH, and SCONPH) demonstrated a strong absorption band at 260 nm, which was attributed to the aromatic amino acids. However, this peak gradually red-shifted and eventually disappeared with the increase of calcium concentration. The similar red shift was also reported on collagen hydrolysate after forming calcium complex [34], which indicated a change in the chiral spatial structure of the hydrolysate chromophores (C=O and -COOH) and co-chromophore groups (-OH and -NH_2_) on the hydrolysate chain after binding with calcium ions. These alterations suggest the formation of hydrolysate–calcium complexes in SCOPH, SCOTH, and SCONPH.

#### 3.2.2. Fluorescence Spectrum of SCOPH, SCOTH, and SCONPH with Calcium

The alterations in the intensity of the fluorescence peak could indicate the interactions between the organic ligands and the metal ions present in the proteins. The fluorescence spectroscopy of samples is shown in Figure 2B_2_. SCOTH showed the strongest fluorescence-quenching effect with the addition of calcium. The fluorescence intensity significantly decreased from 398.62 to 307.50 when the CaCl_2_ concentration increased from 0 M to 1 M. This observation was in agreement with the previous reported study on whey protein hydrolysate–Ca. The reduction in fluorescence intensity with increasing CaCl_2_ was attributed to the quenching effect of metal ions [35]. These findings indicated that the binding of calcium ions to the hydrolysates induces their folding, resulting in the indole group of tryptophan (Trp) shifting from the surface to the interior of the hydrolysate structure.

#### 3.2.3. Effects of Hydrolysate-Calcium Complexes on Chemical Bonding

The modifications in the FTIR absorption peaks of the carboxylates and amides may indicate the interaction between the calcium ions and the hydrolysate (SCOPH, SCOTH, and SCONPH). The amide compounds exhibit characteristic FTIR peaks, including the amide-I band (C=O stretching vibration, 1690–1630 cm^−1^) and amide-II bands (N-H bending (40–60%) and C-N stretching vibrations (18–40%), 1655–1590 cm^−1^) [36]. The FTIR spectra of hydrolysates samples and their calcium complex are shown in Figure 2C_1_–C_3_. After calcium chelation, the amide-I band red-shifted from 1637 cm^−1^, 1632 cm^−1^, and 1606 cm^−1^ to 1647 cm^−1^, 1629 cm^−1^, and 1631 cm^−1^ for SCOPH, SCOTH, and SCONPH, respectively. The amide-II band shifted from 1518 cm^−1^, 1513 cm^−1^, and 1519 cm^−1^ to 1560 cm^−1^, 1561 cm^−1^, and 1563 cm^−1^, for SCOPH, SCOTH, and SCONPH, respectively. Similarly, the N-H bending band at 3386 cm^−1^, 3395 cm^−1^, and 3400 cm^−1^ shifted to 3405 cm^−1^, 3407 cm^−1^, and 3424 cm^−1^ after forming calcium complex for SCOPH, SCOTH, and SCONPH, respectively. These results were consistent with study reported for sea cucumber hydrolysate–calcium chelates [37] and indicate that the chelation of calcium ions by SCOPH, SCOTH, and SCONPH is primarily through the carboxyl oxygen atoms of glutamic acid and aspartic acid, as well as the amino nitrogen atoms.

#### 3.2.4. Particle Size of Hydrolysates and Their Hydrolysates-Calcium Complex

The particle size distributions of SCOPH, SCOTH, and SCONPH and the corresponding calcium complexes (SCOPH-Ca, SCOTH-Ca, and SCONPH-Ca) are shown in Figure 2D. The mean particle sizes of SCOPH, SCOTH, and SCONPH were 113.27 nm, 204.10 nm, and 163.47 nm, respectively, and particle size of their corresponding calcium complex decreased to 103.27 nm, 127.37 nm, and 155.23 nm, respectively.

This decrease in particle size could be attributed to the folding of the hydrolysate structure during calcium chelation, which results in a more compact structure of nanoparticles. These findings agree with the results of the fluorescence spectroscopy, indicating that calcium ions facilitate the folding and aggregation of the hydrolysates. Similarly, Cai et al. [35] reported that the particle size of whey protein hydrolysate decreases upon binding to calcium ions, leading to the formation of dense nanocomposites.

#### 3.2.5. Secondary Structures of SCOPH-Ca, SCOTH-Ca, and SCONPH-Ca

The self-assembly of hydrolysates, induced by metal ion interactions, often results in alterations of their secondary structures. As shown in Figure 2E_1_–E_3_, SCOPH, SCOTH, and SCONPH exhibited a larger negative peak at around 200 nm, while the peak intensity for their calcium complexes was reduced. This indicates the significant alterations in secondary structure during complexes formation.

Compared to pristine hydrolysate, the calcium complexes showed an increase in their β-sheet structure content and a decrease in random coil content. This finding was consistent with results observed for egg white hydrolysate-calcium chelates [38], suggesting that the self-assemble behavior of the hydrolysates may induce the formation of the SCOPH-Ca, SCOTH-Ca, and SCONPH-Ca complexes, with β-sheets playing a dominant role.

As shown in Figure 2F_1_–F_3_, the in vitro simulated gastrointestinal digestion demonstrated a notable reduction in β-sheet content accompanied by an increase in random coil content. Similar results were reported in a study of sea cucumber ovum hydrolysate–calcium chelates by Cui et al. [37], indicating depolymerization of hydrolysate–calcium chelates during gastrointestinal digestion.

#### 3.2.6. Calcium-Chelating Capacity of SCOPH, SCOTH, and SCONPH

The calcium chelation rates of the samples are shown in Figure 2G. Trypsin hydrolysate showed the highest chelation rate of 0.4284 mmol/L among the three different enzymatic digests. The variations in calcium-binding ability among the different types of hydrolysates could be attributed to the difference in their molecular weights, as hydrolysates with molecular weights lower than 3000 Da showed better calcium-binding properties [38]. To enable a deeper understanding, SCOTH was fractionated into hydrolysates smaller than 3000 Da (SHs) and larger than 3000 Da (LHs), and their calcium-binding capacities were compared. The results showed that SHs exhibited higher calcium-binding capacities than LHs, with SHs reaching a value of 0.4714 mmol/L. In a study on the binding interactions between herring egg peptide with calcium, it was reported that the peptide primarily interacted with calcium ions through intermolecular forces like electrostatic interactions [39]. Consequently, SH is a promising biomaterial for calcium chelation.

### 3.3. Inhibitory Effects and Mechanisms of SCOPH, SCOTH, and SCONPH on α-Amylase

#### 3.3.1. Inhibitory Effects of Sea Cucumber Ovum Hydrolysates on α-Amylase Activity

The inhibition rate of samples is shown in Figure 3A. The inhibitory activity against α-amylase increased with rising hydrolysate concentrations, which was consistent with the findings of Nguyen et al. [40], who demonstrated a concentration-dependent inhibitory effect of hydrolysate on α-amylase. The SCOTH exhibited the strongest inhibitory effect, reaching 42%. The sea cucumber ovum hydrolysates with a molecular weight below 3000 Da (SHs) exerted a more pronounced inhibitory effect than the hydrolysates with a molecular weight exceeding 3000 Da (LHs). This aligns with the outcomes of prior research on calcium-ion-binding ability, thereby substantiating the notion that the SH-type hydrolysate is more efficacious.

To evaluate the impact of gastronintestinal digestion on this inhibitory effect, the peptide hydrolysates were injected into the in vitro digestion simulation. The inhibition rate of the sea cucumber ovum protein hydrolysates decreased significantly after enzymatic digestion (using PEPSIN, TYPSIN) (Figure 3B) and may be attributed to the structural degradation of protein hydrolysates by gastrointestinal proteases. The results showed that SCOTH showed the optimum inhibition effect, reaching 42%. Thus, SHs and LHs were separated and purified according to molecular weight. The inhibitory effect of SHs was higher than that of LHs at different molecular weights, which is consistent with the results described earlier for calcium-ion-binding capacity and indicates that SHs were more effective inhibitors than LHs. A comparable consistency has been identified in analogous studies utilizing salmon fish bones, which represent a further marine biological resource [41].

#### 3.3.2. Effects of SCOPH, SCOTH and SCONPH on the Secondary Structure of α-Amylase

The impact of five hydrolysates on the secondary structure of α-amylase was investigated using circular dichroism spectroscopy. As shown in Figure 3C_1_–C_5_, the circular dichroism spectrograms of α-amylase exhibited notable alterations after the addition of SCOPH, SCOTH, and SCONPH. The α-helical content of α-amylase gradually decreased, and β-folded as well as random coiled content increased. This indicates that SCOPH, SCOTH, and SCONPH may cause conformational changes in α-amylase and reduce the active sites of α-amylase, thus inhibiting enzyme activity. Similarly, Lu et al. [30] reported that the interaction of α-amylase with its inhibitor caused alteration in the secondary structure.

#### 3.3.3. Effect of SCOPH, SCOTH and SCONPH on the Particle Size of α-Amylase

The hydrodynamic radius of α-amylase was significantly impacted by the addition of SCOPH, SCOTH, and SCONPH, as demonstrated by Figure 3D_1_–D_5_. Without SCOPH, SCOTH, and SCONPH, the hydrodynamic radius of α-amylase, a key parameter in understanding the enzyme’s behavior, was 495.23 ± 4.34 nm. With the gradual increase in the concentration of the added hydrolysate, the hydrodynamic radius of α-amylase also increased. This could be ascribed to the function of the hydrolysate as the nucleus for the aggregation of α-amylase, which results in the formation of larger copolymers derived from the initial small aggregates. The incorporation of the hydrolysate resulted in a modification of the aggregation process of α-amylase.

#### 3.3.4. Interaction of SCOPH, SCOTH, SCONPH, SHs, and LHs with α-Amylase

Fluorescence spectroscopy represents a methodology for the investigation of the interaction between small-molecule compounds and proteins. As shown in Figure 4A_1_–A_3_–E_1_–E_3_, the fluorescence quenching effect of hydrolysates on α-amylase was observed, with a gradual decline in intensity with the addition of hydrolysate. This could be attributed to the complex formed between hydrolysate and enzyme. Similarly, Lu et al. [30] found that acteoside can interact with α-amylase, leading to a reduction in fluorescence. Since fluorescence quenching is temperature-dependent, the Stern–Volmer equation at three different temperatures (298, 308, and 318 K) was used to confirm the quenching mechanism between hydrolysates and α-amylase.

The quenching constants between SCOPH, SCOTH, SCONPH, and α-amylase are presented in Table 1. Figure 4A_4_–E_4_ and Table 1 show that the Stern–Volmer equation for SCOPH demonstrated a linear relationship, and the K_SV_ value exhibited a decline with rising temperature, indicating that the quenching of SCOPH, SCOTH, and SCONPH with α-amylase was of the static quenching type.

### 3.4. Mode and Type of Inhibition

In Figure 5A_1_–A_5_, the inhibitory mechanism of hydrolysates on α-amylase is represented by the convergence of all lines at the origin. As the concentration of the inhibitor increases, the slope of the lines in question decreases. According to Lineweaver–Burk plot, the inhibition of enzymes can be classified into four categories: mixed, anticompetitive, non-competitive, and competitive. The type of inhibition can be determined by identifying the point of intersection of the lines on this plot [42].

Similarly, Silva et al. [43] observed that amino acid derivatives exerted a concentration-dependent inhibitory effect on α-amylase activity. This indicates that the inhibitory effect of SCOPH, SCOTH, and SCONPH on α-amylase is likely to be reversible. The slope for SHs demonstrated the most substantial decline, and the straight line for SHs was the closest to the horizontal axis at the highest inhibitor concentration, indicating that SHs exhibited the most pronounced inhibition of α-amylase.

In Figure 5B_1_–B_5_, the straight lines intersect in the second quadrant, suggesting that the inhibition of α-amylase by SCOPH, SCOTH, and SCONPH can be classified as mixed inhibition. In other words, the hydrolysates could either bind to the active site of the enzyme, performing competitive inhibition, or bind to non-active sites, exerting non-competitive inhibition. The inhibition constants (K_I_ and K_IS_) of SCOPH, SCOTH, and SCONPH on α-amylase were calculated using Lineweaver–Burk double reciprocal plots. As shown in Table 2, the K_IS_ values for SCOPH, SCOTH, and SCONPH were greater than their corresponding K_I_ values and SHs exhibited the smallest inhibition constants (K_I_ and K_IS_), indicating that SHs inhibited α-amylase more effectively.

## 4. Conclusions

The inhibition of α-amylase through interaction with dietary small molecules is generally considered a safe and efficacious method for obesity prevention and management. The results from this study indicated that all the sea cucumber ovum protein hydrolysates exhibited promising calcium-chelating properties, with the trypsin hydrolysate demonstrating the most effective inhibitory effect on α-amylase activity. The sea cucumber ovum trypsin hydrolysate was purified to yield SHs with a molecular weight below 3000 Da. It showed a stronger affinity for calcium ions and a concentration-dependent mixed-inhibition effect on α-amylase. The results also indicate trypsin hydrolysate, with an inhibitory effect on digestive enzymes, is a potential candidate for a food-derived weight management supplement. This study demonstrated the feasibility of utilizing marine by-products as health promotion ingredients in an added-value approach.

## Figures and Tables

**Figure 1 foods-13-04119-f001:**
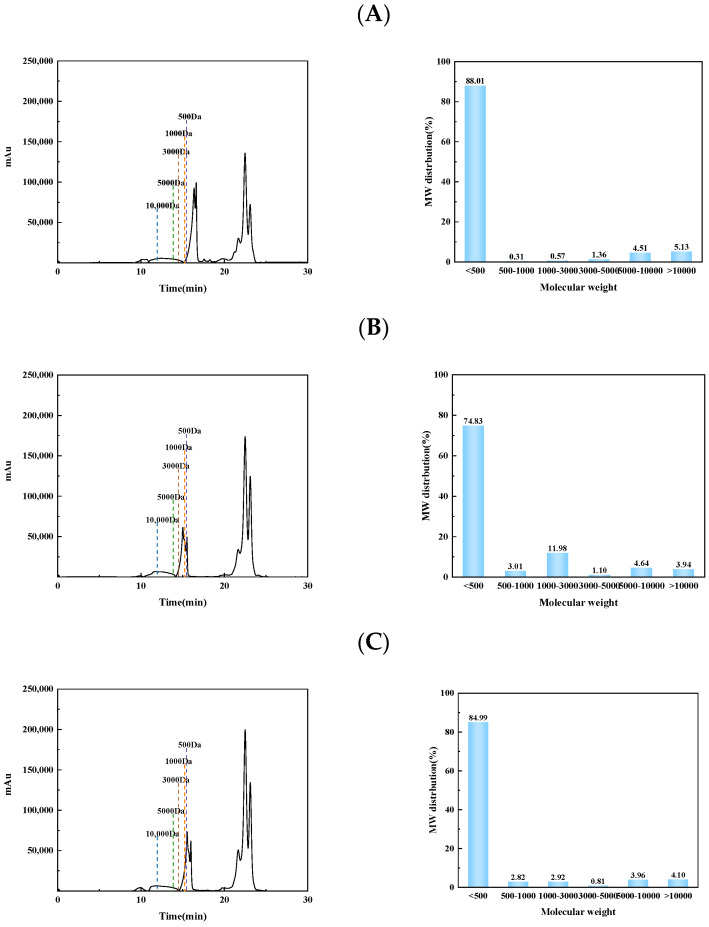
Molecular weight distribution curve and percentage of sea cucumber ovum hydrolysate: (**A**) SCOPH, (**B**) SCOTH, and (**C**) SCONPH.

**Figure 2 foods-13-04119-f002:**
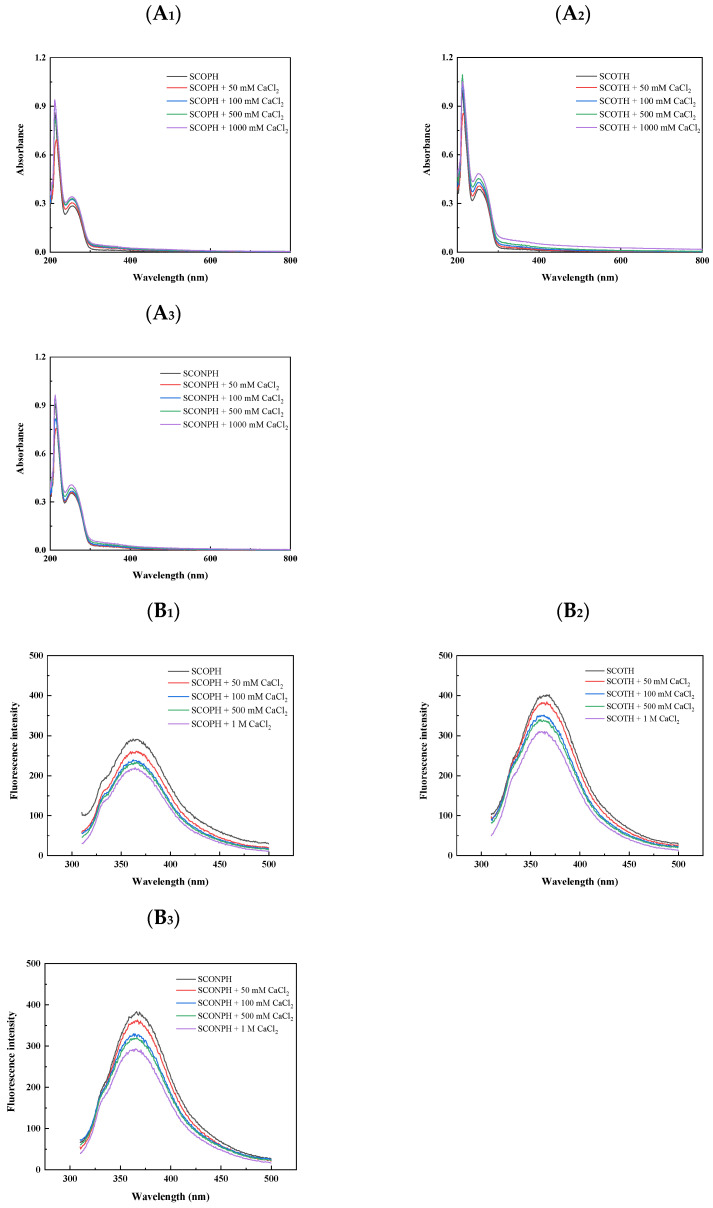
Preparation and properties of calcium chelate from sea cucumber ovum hydrolysate. (**A_1_**–**A_3_**): UV absorption spectra analysis of hydrolysates and corresponding hydrolysate-calcium complexes obtained by hydrolysis of sea cucumber ovum with three enzymes in the 190–800 nm range: (**A_1_**) SCOPH, (**A_2_**) SCOTH, and (**A_3_**) SCONPH. (**B_1_**–**B_3_**): Fluorescence spectra of the hydrolysates obtained from the hydrolysis of sea cucumber ovum by three enzymes and different concentrations of CaCl_2_ in the excitation wavelength range of 295 nm and emission wavelength of 310–500 nm: (**B_1_**) SCOPH, (**B_2_**) SCOTH, and (**B_3_**) SCONPH. (**C_1_**–**C_3_**): FT-IR analysis of the hydrolysate and the corresponding hydrolysate–calcium complexes obtained from the hydrolysis of sea cucumber ovum by three enzymes in the range of 4000–400 cm^−1^: (**C_1_**) SCOPH, (**C_2_**) SCOTH, and (**C_3_**) SCONPH. (**D**): Particle size variation in sea cucumber ovum hydrolysates and their calcium chelates. (**E_1_**–**E_3_**): Circular dichroism spectra and secondary structure content of sea cucumber ovum hydrolysates and their calcium complexes: (**E_1_**) SCOPH, (**E_2_**) SCOTH, and (**E_3_**) SCONPH. (**F_1_**–**F_3_**): Changes in the secondary structure of sea cucumber ovum hydrolysates and their calcium complexes during simulated gastro ovum digestion: (**F_1_**) SCOPH, (**F_2_**) SCOTH, and (**F_3_**) SCONPH. (**G**): Calcium-binding capacity of hydrolysates obtained by enzymatic digestion of sea cucumber ovum by different proteases. Different letters indicate significant differences (*p* < 0.05).

**Figure 3 foods-13-04119-f003:**
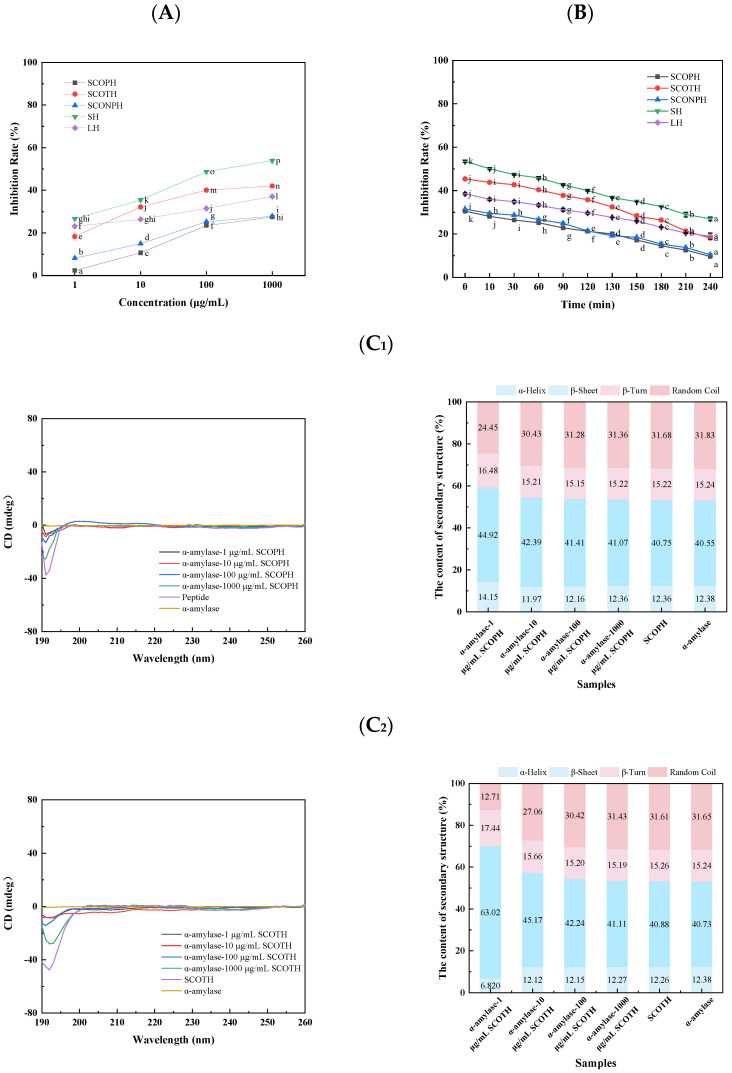
Effect of sea cucumber ovum hydrolysate on α-amylase inhibitory activity. (**A**) The change in inhibition rate with peptide concentration. (**B**) Changes in inhibition rate during simulated digestion. (**C_1_**–**C_3_**) Effect of different protease hydrolysate concentrations on the secondary structure content of α-amylase: (**C_1_**) SCOPH, (**C_2_**) SCOTH, (**C_3_**) SCONPH, (**C_4_**) SH, and (**C_5_**) LH. (**D_1_**–**D_3_**) Effect of hydrolysate concentrations of different protease hydrolysates on α-amylase particle size: (**D_1_**) SCOPH, (**D_2_**) SCOTH, (**D_3_**) SCONPH, (**D_4_**) SH, and (**D_5_**) LH. Different letters indicate significant differences (*p* < 0.05).

**Figure 4 foods-13-04119-f004:**
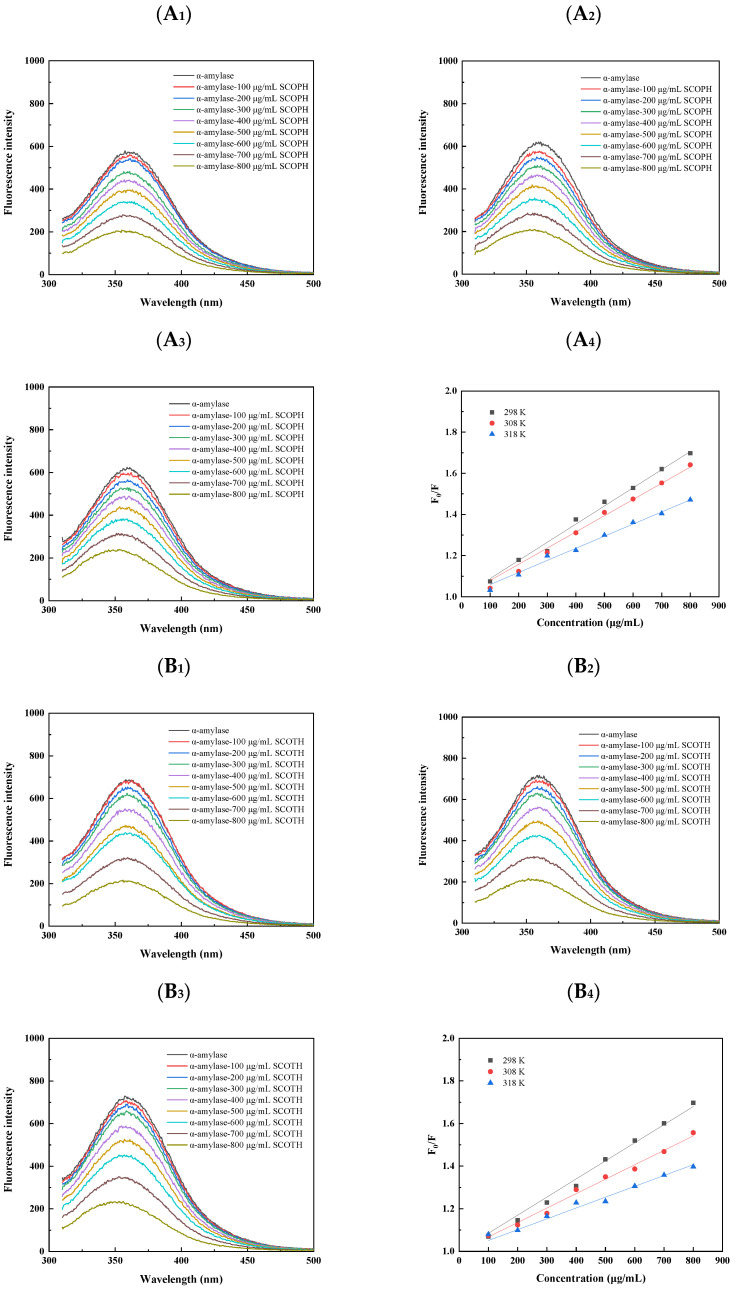
Fluorescence spectra of α-amylase with sea cucumber ovum digests at different concentrations: (**A_1_**,**B_1_**,**C_1_**,**D_1_**,**E_1_**) 298 K, (**A_2_**,**B_2_**,**C_2_**,**D_2_**,**E_2_**) 308 K, (**A_3_**,**B_3_**,**C_3_**,**D_3_**,**E_3_**) 318 K, and (**A_4_**,**B_4_**,**C_4_**,**D_4_**,**E_4_**) Stern–Volmer of sea cucumber ovum digests inducing intrinsic fluorescence quenching of α-amylase at different temperatures.

**Figure 5 foods-13-04119-f005:**
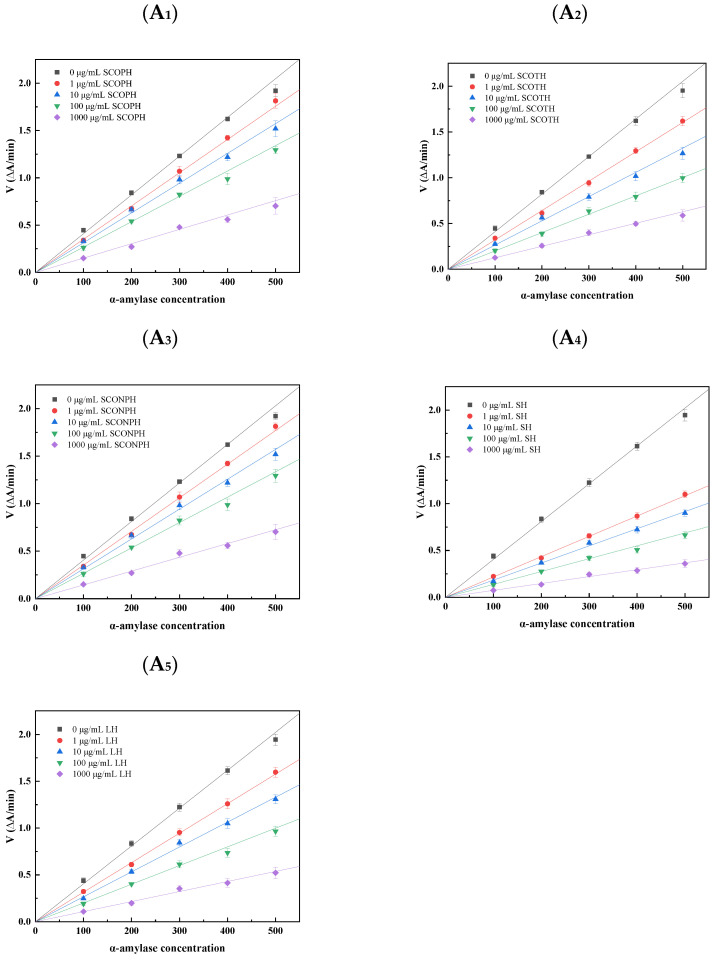
Mode and type of inhibition. (**A_1_**–**A_5_**) Analysis of the inhibitory effect of sea cucumber ovum hydrolysate on α–amylase. (**B_1_**–**B_5_**) Lineweaver–Burk curves of sea cucumber ovum hydrolysate on α–amylase.

**Table 1 foods-13-04119-t001:** Quenching constants of sea cucumber ovum hydrolysates with α-amylase.

λ_ex_ = 280 nm	T (K)	Equation	R^2^	K_sv_ (1 × 10^3^ L/g)	K_q_ (1 × 10^11^ L/g∙s)
SCOPH	298	y = 9.019x + 0.989	0.991	8.82 ± 1.46	8.82 ± 1.46
	308	y = 8.602x + 0.959	0.997	7.88 ± 1.55	7.88 ± 1.55
	318	y = 6.104x + 0.988	0.991	5.89 ± 1.07	5.89 ± 1.07
SCOTH	298	y = 9.131x + 0.964	0.997	8.50 ± 1.48	8.50 ± 1.48
	308	y = 6.878x + 0.994	0.991	6.77 ± 1.16	6.77 ± 1.16
	318	y = 4.704x + 1.021	0.986	5.09 ± 1.20	5.09 ± 1.20
SCONPH	298	y = 9.039x + 0.939	0.992	7.97 ± 2.50	7.97 ± 2.50
	308	y = 8.660x + 0.947	0.999	7.74 ± 1.86	7.74 ± 1.86
	318	y = 6.237x + 0.960	0.991	5.53 ± 1.69	5.53 ± 1.69
SH	298	y = 8.807x + 1.002	0.985	8.85 ± 1.70	8.85 ± 1.70
	308	y = 6.690x + 0.985	0.986	6.63 ± 1.37	6.63 ± 1.37
	318	y = 4.841x + 0.957	0.986	4.08 ± 1.73	4.08 ± 1.73
LH	298	y = 8.648x + 0.983	0.989	8.35 ± 1.57	8.35 ± 1.57
	308	y = 6.795x + 0.969	0.995	6.25 ± 1.29	6.25 ± 1.29
	318	y = 4.401x + 0.970	0.984	3.87 ± 1.36	3.87 ± 1.36

**Table 2 foods-13-04119-t002:** Inhibition constants and type of inhibition of α-amylase by sea cucumber ovum hydrolysates.

Enzyme	Sample	K_I_ (mg/mL)	K_IS_ (mg/mL)	Inhibition Type
α-amylase	SCOPH	6.79	8.19	Mixed inhibition
	SCOTH	4.53	5.45	Mixed inhibition
	SCONPH	5.53	6.65	Mixed inhibition
	SH	4.67	5.64	Mixed inhibition
	LH	5.87	7.07	Mixed inhibition

## Data Availability

The original contributions of the study are included in the article. Further inquiries can be directed to the corresponding authors.

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
