# Peer review of "Preparation and Characterization of Calcium-Chelated Sea Cucumber Ovum Hydrolysate and the Inhibitory Effect on α-Amylase"

_foods, 2024, doi:10.3390/foods13244119_

Round 1
Reviewer 1 Report
Comments and Suggestions for Authors
Abstract has to be modified . Results can be written clearly with minimal figures. discussion has to be improved .
Comments on the Quality of English Languageabstract and results have to be improved .
Author Response
Reviewer #1:
Question : Abstract has to be modified. Results can be written clearly with minimal figures. discussion has to be improved. Abstract and results have to be improved.
Response: Thanks for the helpful comments. It has been revised in the revised manuscript. The results of the study have been presented in a more coherent manner, and the use of numerical data has been minimized. Furthermore, the discussion section has been augmented with the objective of enhancing the paper's overall quality. The guidance is greatly appreciated, and these revisions have been implemented. Manuscript has undergone English language editing by MDPI-english-87724.

Reviewer 2 Report
Comments and Suggestions for Authors
ch as seafood, have considerable potential for incorporation into functional foods andThis study investigates sea cucumber ovum hydrolysates, highlighting their capacity to inhibit α-amylase and reduce energy absorption. Three hydrolysates were produced using different proteases (pepsin, trypsin, and neutral protease), with the trypsin-derived hydrolysate under 3000 Da showing the highest inhibition rate (53.9%). The hydrolysates also demonstrated strong calcium-chelating capacity and a fluorescence burst effect on α-amylase. With these enhancements, the article should be accepted, as it contributes to the sustainable use of seafood by-products.
This study investigates sea cucumber ovum hydrolysates, highlighting their capacity to inhibit α-amylase and reduce energy absorption. Three hydrolysates were produced using different proteases (pepsin, trypsin, and neutral protease), with the trypsin-derived hydrolysate under 3000 Da showing the highest inhibition rate (53.9%). The hydrolysates also demonstrated strong calcium-chelating capacity and a fluorescence burst effect on α-amylase. With these enhancements, the article should be accepted, as it contributes to the sustainable use of seafood by-products.
1- “in contrast, bioactive peptides derived from natural sources show distinct advantages, including improved safety, digestibility, and reduced irritation”
How does the molecular weight of hydrolysates affect their α-amylase inhibitory activity?
2- “It can be reasonably deduced that edible aquatic animals, particularly by-products such as seafood…”
Introduce the concept of reusing parts of aquatic beings, such as fish proteases ( https://doi.org/10.1016/j.bcab.2020.101584)
3- “The sea cucumber ovum hydrolysates generated using pepsin, trypsin, and protease were labeled as SCOPH, SCOTH, and SCONPH respectively.”
What are the distinct roles of pepsin, trypsin, and neutral protease in the bioactivity of the hydrolysates?
4- “The molecular weight of the hydrolysates was a significant factor in their calcium ion chelating ability, with hydrolysates below 3000 Da typically showing a higher affinity to calcium ions”
Is there any evidence for structural specificity of peptides in calcium binding?
5- How consistent is the α-amylase inhibition rate of 53.9% with similar studies using other marine bioresources?
6- Did the study examine potential cytotoxicity or safety of the hydrolysates in a biological setting?other dietary interventions
In contrast, bioactive peptides derived from natural sources show distinct ad-
Author Response
Reviewer #2:
Question 1 : “in contrast, bioactive peptides derived from natural sources show distinct advantages, including improved safety, digestibility, and reduced irritation”
How does the molecular weight of hydrolysates affect their α-amylase inhibitory activity?
Response: Thanks for the helpful comments. The α-amylase inhibitory activity is significantly affected by the molecular weight of the hydrolysate. In general, better inhibitory activity is had by lower molecular weight hydrolysates (e.g., low molecular weight peptides and amino acids) because the active site of the enzyme is more likely to be bound by them and the substrate is prevented from entering. The enzyme activity may not be effectively inhibited by higher molecular weight hydrolysates due to their large size. The inhibitory ability is also affected by the specific amino acid composition, and within an optimized molecular weight range, stronger inhibitory activity is exhibited by certain amino acids. In addition, the solubility and stability of the hydrolysate can be affected by a higher molecular weight, thereby its bioavailability and inhibitory effect being affected. Overall, an important role is played by molecular weight and structural properties in α-amylase inhibitory activity.
References cited
Liu, C., Ding, W.-j., Huo, Y., & Liu, A.-j. (2023). Comprehensive assessment of peptide derived from pig spleen: Prepara-tion, bioactivity and structure-activity relationships. Food Bioscience, 56.
Sommella, E., Pepe, G., Ventre, G., Pagano, F., Conte, G. M., Ostacolo, C., Manfra, M., Tenore, G. C., Russo, M., Novellino, E., & Campiglia, P. (2016). Detailed peptide profiling of “Scotta”: from a dairy waste to a source of potential health-promoting compounds. Dairy Science & Technology, 96(5), 763-771.
Question 2 : “It can be reasonably deduced that edible aquatic animals, particularly by-products such as seafood…”
Introduce the concept of reusing parts of aquatic beings, such as fish proteases ( https://doi.org/10.1016/j.bcab.2020.101584)
Response: Thanks for the helpful comments. Considerable attention has been garnered by the reuse of aquatic animals, particularly by-products such as seafood, due to its potential to enhance the sustainability of the food industry and reduce waste. One noteworthy aspect is the utilization of fish proteases, enzymes extracted from fish that facilitate the breakdown of proteins into smaller peptides and amino acids. Significant applications in diverse fields, including food processing, medicine, and biotechnology, are possessed by these proteases. The potential to enhance the nutritional quality of products, as well as to harness their functional properties, including the improvement of flavor, the enhancement of digestibility, and the extension of shelf life, is had by the reintroduction of fish proteases into the food system. Moreover, consistency with the tenets of the circular economy, which advocates environmental sustainability through the minimisation of waste and the reuse of resources, is shown by the utilisation of these by-products. For instance, the pivotal role in diverse biotechnological applications has been underscored by recent studies of fish-derived proteases, underscoring their value as a resource beyond their original use. The optimization of the value of aquatic resources is achieved by this approach, and the development of more sustainable food systems that utilize every component of the organism is also fostered, thereby the environmental impact being reduced.
References cited
Ribeiro Cardoso dos Santos, D. M., Victor dos Santos, C. W., Barros de Souza, C., Sarmento de Albuquerque, F., Marcos dos Santos Oliveira, J., & Vieira Pereira, H. J. (2020). Trypsin purified from Coryphaena hippurus (common dolphinfish): Purification, characterization, and application in commercial detergents. Biocatalysis and Agricultural Biotechnology, 25.
Question 3 : “The sea cucumber ovum hydrolysates generated using pepsin, trypsin, and protease were labeled as SCOPH, SCOTH, and SCONPH respectively.”
What are the distinct roles of pepsin, trypsin, and neutral protease in the bioactivity of the hydrolysates?
Response: Thanks for the helpful comments. An acidic environment is where pepsin is active, and the amide bonds of amino acids such as phenylalanine and tryptophan are primarily hydrolyzed by it, with the result that highly biologically active hydrolyzates that are suitable for initial digestion are produced. In contrast, trypsin functions optimally in neutral or slightly alkaline conditions, and lysine and arginine are specifically hydrolyzed by it. Favorable digestive and absorptive properties are exhibited by the resulting hydrolysates, which are thus rendered suitable for further digestion.Neutral proteases are widely distributed in various tissues and a wide range of substrate specificities are exhibited by them. They primarily function under neutral conditions. The biological activity and nutritional value of the resulting hydrolysates are contingent upon the specific protein type.In conclusion, a crucial role in the digestive process and the enhancement of nutritional value is played by these three proteases.
References cited
Duan, Y., Deng, D., Yang, X., Zhang, L., Ma, X., He, L., Ma, G., Li, S., & Li, H. (2024). Bovine liver hydrolysates based on six proteases: Physicochemical properties, emulsification characteristics, antioxidant capacity assessment, and peptide identification. Lwt, 208.
Question 4 : “The molecular weight of the hydrolysates was a significant factor in their calcium ion chelating ability, with hydrolysates below 3000 Da typically showing a higher affinity to calcium ions.”
Is there any evidence for structural specificity of peptides in calcium binding?
Response: Thanks for the helpful comments. It is believed that the capacity of the peptide to bind calcium ions is crucially influenced by its structural specificity. Electrostatic interactions with calcium ions can be facilitated by the presence of specific amino acids, including glutamic acid, aspartic acid, and histidine. Additionally, the formation of accessible binding sites for calcium binding can be contributed to by the secondary structure of the peptide, such as an α-helix or β-fold. In general, high calcium ion affinity is exhibited by small peptides with a molecular weight below 3000 Da due to the high density of binding sites and the relatively small molecular volume. Moreover, the formation of a stable coordination complex between the functional groups of specific amino acids in the peptide and the calcium ion is typically entailed in the process of calcium binding. It has been demonstrated by studies that specific sequences that enhance their binding efficiency are exhibited by calcium-binding peptides derived from diverse sources, including dairy products and marine organisms. Collectively, the capacity of peptides to bind calcium ions is markedly influenced by their structural specificity.
References cited
Sun, N., Wang, Y., Bao, Z., Cui, P., Wang, S., & Lin, S. (2020). Calcium binding to herring egg phosphopeptides: Binding characteristics, conformational structure and intermolecular forces. Food Chem, 310, 125867.
Question 5 : How consistent is the α-amylase inhibition rate of 53.9% with similar studies using other marine bioresources?
Response: Thanks for the helpful comments. The α-amylase inhibition rate of 53.9% is relatively consistent with similar studies on marine biological resources, but significant variability is also demonstrated depending on a number of factors. Different inhibitory activities are displayed by different marine organisms (such as fish, shellfish, seaweed, etc.) depending on the biological source and extraction method. In addition, the properties of the peptides can be affected by differences in extraction and hydrolysis techniques (such as the use of different enzymes or extraction solvents), with their inhibitory effect on α-amylase being altered as a result. An important role in their inhibitory capacity is played by the size and amino acid composition of the peptides. Stronger inhibitory potential is generally exhibited by smaller peptides or peptides rich in specific amino acids (e.g. aromatic residues). Enzyme activity and the interaction between α-amylase and inhibitory peptides can also be influenced by differences in experimental conditions (such as pH, temperature, and duration of the experiment). Inhibition rates as high as 53.9% have been reported in some studies of marine origin, and consistency with the results of this study is shown since hydrolysates of certain fish and shellfish typically range from 50% to 60%.
References cited
Xu, Z., Han, S., Cui, N., Liu, H., Yan, X., Chen, H., Wu, J., Tan, Z., Du, M., & Li, T. (2024). Identification and characterization of a calcium-binding peptide from salmon bone for the targeted inhibition of alpha-amylase in digestion. Food Chem X, 22, 101352.

Reviewer 3 Report
Comments and Suggestions for Authors
This is a consistent report focused on the inhibition effect caused by three hydrolysates of the see cucumber ovum protein on the alpha - amylase activity. It was found that pepsin, trypsin and neutral protease are usually encountered in marine echinoderm intestines which are removed as waste. However, these compounds could have significant inhibitory impact on calcium binding peptides extracted from see cucumber ovum.
In this context, a careful analysis has been proposed of such hydrolysates and their calcium complexes by applying consecrated analytical methods: molecular weight distribution, UV-VIS, FT-IR and fluorescence spectroscopies, particle size distribution, circular dichroism, calcium binding capacity, and inhibition mechanism.
The section devoted to Results and Discussion is largely furnished in detail with figures, even though it would be preferable to be quantified in more tables.
By elucidating the inhibition mechanism of these calcium complexes this study might bring valuable contribution on the obtaining of healthy nutritive additives, particularly as a solution against obesity.
The volume of the paper is big enough to justify the authors number (10). However, the personal contribution of each of them should be clearly mentioned at the end of the text.
Author Response
Reviewer #3:
Question : This is a consistent report focused on the inhibition effect caused by three hydrolysates of the see cucumber ovum protein on the alpha - amylase activity. It was found that pepsin, trypsin and neutral protease are usually encountered in marine echinoderm intestines which are removed as waste. However, these compounds could have significant inhibitory impact on calcium binding peptides extracted from see cucumber ovum.
In this context, a careful analysis has been proposed of such hydrolysates and their calcium complexes by applying consecrated analytical methods: molecular weight distribution, UV-VIS, FT-IR and fluorescence spectroscopies, particle size distribution, circular dichroism, calcium binding capacity, and inhibition mechanism.
The section devoted to Results and Discussion is largely furnished in detail with figures, even though it would be preferable to be quantified in more tables.
By elucidating the inhibition mechanism of these calcium complexes this study might bring valuable contribution on the obtaining of healthy nutritive additives, particularly as a solution against obesity.
The volume of the paper is big enough to justify the authors number (10). However, the personal contribution of each of them should be clearly mentioned at the end of the text.
Response: Thanks for the helpful comments. In response to the recommendations outlined in the "Results and Discussion" section, the manuscript has been meticulously revised to present the information in a more comprehensive and accessible manner. The individual contributions of each author have been explicitly delineated in the CRediT authorship contribution statement to ensure that their respective work is duly acknowledged. The manuscript has been edited in English by MDPI-english-87724.
